# VFMStitch: A Vision-Foundation-Model Empowered Framework for 3D Ultrasound Stitching via Geometric–Semantic Feature Fusion

**Xing Yao**[1]                                    XING.YAO@VANDERBILT.EDU
**Nick DiSanto**[1]                           NICOLAS.C.DISANTO@VANDERBILT.EDU
**Runxuan Yu**[1]                            RUNXUAN.YU@VANDERBILT.EDU
**Jiacheng Wang**[1]                    JIACHENG.WANG.1@VANDERBILT.EDU
**Daiwei Lu**[1]                                DAIWEI.LU@VANDERBILT.EDU
**Gabriel Arenas**[2]                GABRIEL.ARENAS@PENNMEDICINE.UPENN.EDU
**Baris Oguz**[2]                                 BARISUMOG@GMAIL.COM
**Alison Pouch**[2]                         POUCH@PENNMEDICINE.UPENN.EDU
**Nadav Schwartz**[2]            NADAV.SCHWARTZ@PENNMEDICINE.UPENN.EDU
**Brett C Byram**[1]                       BRETT.C.BYRAM@VANDERBILT.EDU
**Ipek Oguz**[1]                                IPEK.OGUZ@VANDERBILT.EDU
[1] *Vanderbilt University*
[2] *University of Pennsylvania*

**Editors:** Accepted for publication at MIDL 2026

## Abstract

3D ultrasound (3DUS) stitching expands the field-of-view (FOV) by registering partially overlapping 3DUS volumes acquired from different probe positions. This task is intrinsically difficult due to large inter-volume translations and rotations, the impact of the sector-shaped FOV, as well as the heavy noise and artifacts inherent to ultrasound. With the rapid progress of Vision Foundation Models (VFMs) such as DINOv3, VFM-derived features have recently shown promise for downstream medical image registration tasks. However, existing VFM-based approaches primarily focus on deformable registration and are rarely evaluated for rigid alignment under large motions. Moreover, the feasibility of leveraging VFM-derived features for robust 3DUS stitching remains largely unexplored. In this study, we introduce **VFMStitch**, the first training-free, VFM-empowered 3DUS stitching framework that integrates point-cloud (PCD)–based geometric features with DINOv3-derived semantic descriptors. Extensive experiments demonstrate that VFMStitch substantially improves rigid registration accuracy compared to existing methods, validating the effectiveness of geometric–semantic fusion for challenging 3DUS stitching scenarios. The code is available at github.com/MedICL-VU/VFMStitch.

**Keywords:** vision foundation model, DINOv3, ultrasound, stitching, point cloud, feature fusion

## 1. Introduction

Ultrasound (US) image registration (Che et al., 2017; Entrekin et al., 2001; Wang et al., 2014) is a pivotal task to many downstream analysis tasks, with US stitching (Banerjee et al., 2015; Gomez et al., 2019; Wright et al., 2023; Bano and Stoyanov, 2024) being a

key application for expanding the field of view (FOV) by aligning partially overlapping scans from different probe positions. This is particularly important for visualizing large anatomical structures such as the fetus and placenta (Roy-Lacroix et al., 2017; Gomez et al., 2017). However, large inter-volume translations and rotations frequently arise during freehand scanning (Yao et al., 2025b), making rigid alignment for 3DUS stitching inherently difficult. Furthermore, the effect of the sector-shaped FOV (Yao et al., 2025a), low signal-to-noise ratio and artifacts of US imaging (Yao et al., 2024) weaken the reliability of intensity-based similarity measures and reduce the distinctiveness of local features. These factors collectively impose significant limitations on intensity-based or feature-based registration pipelines and motivate the exploration of more robust, representation-driven solutions.

To address these challenges, prior studies have explored a variety of strategies, including block-matching (Banerjee et al., 2015), manifold-learning–based keypoint selection (Gomez et al., 2019, 2017), iterative spatial transformer network (Wright et al., 2019), and reinforcement learning (Wright et al., 2023). More recently, diffusion-based frameworks such as SynStitch (Yao et al., 2025b) and LOTUS (Yao et al., 2025a) have been proposed to explicitly mitigate the impact of the sector-shaped FOV in both 2D and 3D US stitching scenarios. While these approaches represent meaningful advances, they often require large-scale training data and remain sensitive to variations in image quality. A training-free, robust, and accurate 3DUS stitching framework is thus desirable.

Recently, self-supervised vision foundation models (VFMs), such as the DINO family (Caron et al., 2021; Oquab et al., 2023; Siméoni et al., 2025), have attracted substantial attention due to their strong generalization ability and competitive zero-shot performance on a wide range of medical imaging tasks (Ambsdorf et al., 2025; Li et al., 2025; Avants et al., 2008; Wang et al., 2025; Gu et al., 2025). Among these efforts, DINO-Reg (Avants et al., 2008) and DINOv3+T$^3$ (Wang et al., 2025) have demonstrated state-of-the-art results in deformable medical image registration by leveraging DINOv2 (Oquab et al., 2023) and DINOv3 (Siméoni et al., 2025) features in combination with test-time optimization strategies (Siebert et al., 2024). Despite this promising progress, significant gaps remain. First, existing VFM-based registration frameworks primarily target deformable alignment, leaving rigid registration under large translations and rotations insufficiently explored. Second, prior evaluations have been conducted largely on high-quality modalities such as MRI and CT, where texture and contrast are substantially more stable. Their performance in artifact-prone, and highly heterogeneous 3D ultrasound, particularly in demanding scenarios such as placenta imaging, remains an open question. These limitations motivate the exploration of how VFMs can be adapted or extended to address the unique difficulties of 3DUS stitching.

In this work, we propose **VFMStitch**, the first training-free, VFM-empowered framework specifically designed for 3DUS stitching under large rigid motions by performing point-cloud (PCD)–based registration on the fusion of PCD-based geometric features with DINOv3-derived semantic descriptors. Both qualitative and quantitative evaluations show that VFMStitch significantly outperforms state-of-the-art methods in challenging 3DUS stitching scenarios. Our main contributions are as follows:

- **Methodological novelty:** VFMStitch is the first framework to leverage VFM-derived features for medical image registration involving *large rigid transformations*, filling a critical gap left by existing deformable-focused VFM approaches.

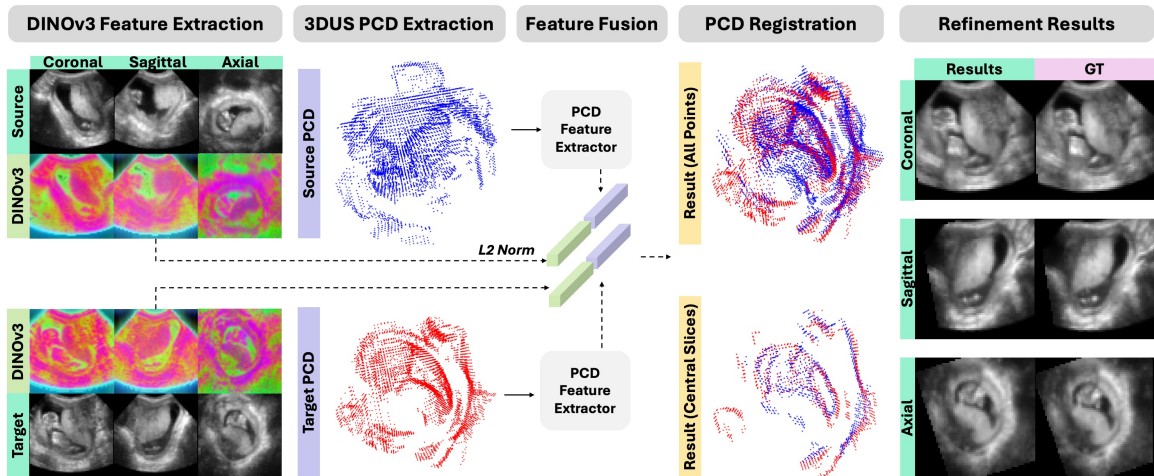

Figure 1: Pipeline of VFMStitch. DINOv3-derived semantic features are extracted from the stitching pair, followed by 3DUS PCD extraction and geometric descriptor computation. Semantic and geometric features are fused for PCD-based registration, and the result is refined via unsupervised intensity-based registration.

- **Geometric robustness:** We show that PCD-based registration serves as a reliable and robust alternative to intensity-based registration for 3DUS stitching, particularly in the presence of strong noise and artifacts.

- **Semantic descriptor superiority:** We demonstrate that DINOv3-based semantic features significantly outperform traditional and learning-based descriptors when used as PCD registration features.

- **Effectiveness of geometric-semantic feature fusion:** Our results further reveal that fusing geometric and semantic cues yields additional performance gains, highlighting the complementary nature of these representations.

## 2. Methods

Fig. 1 illustrates the VFMStitch pipeline. A stitching pair $(I_{source}, I_{target})$ refers to two partially overlapping 3DUS volumes acquired from the same subject at different probe positions. For each stitching pair, DINOv3-derived semantic features are first extracted, with the top three principal components visualized in RGB via principal component analysis (PCA). Point clouds (PCDs) are then extracted from the corresponding gray-scale volumes, and geometric descriptors are computed. The semantic and geometric features are fused to perform PCD-based registration from the source to target image. Finally, the estimated transformation is propagated to voxel space and refined through unsupervised intensity-based registration.

## 2.1. Unsupervised 3DUS PCD Extraction

As intensity-based registration struggles under heterogeneous image quality, geometric alternatives based on PCDs have been explored. In ultrasound applications, PCDs are often extracted from segmentation masks (Jiang et al., 2023, 2024; Tan et al., 2023), which is suboptimal for 3DUS stitching: manual annotation is costly, automated segmentation generalizes poorly, and object-centric masks may discard contextual anatomy when inter-volume overlap is limited, resulting in poor registration.

To address these limitations, we propose an unsupervised, segmentation-free PCD extraction strategy based on edge representations that preserve rich anatomical geometry. Specifically, edge maps are computed using complementary classical operators, including the Sobel filter (Kanopoulos et al., 1988), Laplacian of Gaussian (LoG) (Marr and Hildreth, 1980), and Harris corner detector (Harris et al., 1988), capturing intensity gradients, fine structural ridges, and stable corner-like features, respectively. Field-of-view (FOV) masks are applied to suppress boundary artifacts induced by the sector-shaped 3DUS geometry, retaining only anatomically relevant regions. The filtered edge maps are then averaged and normalized to the range $[0, 1]$ to form a unified geometric representation. Compared to intensity- or segmentation-based cues, this edge-based representation provides a more complete and noise-robust characterization of 3DUS anatomy for registration initialization.

For a given 3DUS volume $I \in \mathbb{R}^{D \times H \times M}$, the final 3DUS point cloud is denoted as $\{\mathbf{p}_i\}_{i=1}^N$, where $\mathbf{p}_i \in \mathbb{R}^3$. $\{\mathbf{p}_i\}_{i=1}^N$ is extracted by thresholding the merged edge map and subsequently the number of points is reduced by $1/r$ using Open3D (Zhou et al., 2018), where $r = 5$ is the default downsampling ratio in our experiments.

## 2.2. Geometric-Semantic Feature Fusion

**PCD-based geometric descriptor extraction:** To extract a local geometric descriptor $\mathbf{f}^{\text{Geo}}i \in \mathbb{R}^{d\text{Geo}}$ for each point $\mathbf{p}_i$, we investigate a diverse set of PCD geometric descriptors, including both conventional hand-crafted methods and state-of-the-art learning-based approaches. Specifically, we evaluate Fast Point Feature Histograms (FPFH) (Rusu et al., 2009) as a classical baseline, as well as recent learning-based models, including Point-MAE (Pang et al., 2022), Point-DAE (Zhang et al., 2025), and Point-BERT (Yu et al., 2022). For FPFH, normal vectors are estimated via KD-tree search (radius: 10, up to 30 neighbors), followed by descriptor extraction with a larger neighborhood (radius: 25, up to 100 neighbors), yielding features of dimensionality $d_{\text{Geo}} = 33$. Learning-based descriptors are extracted using pretrained models without fine-tuning, producing per-point features with a unified dimensionality of $d_{\text{Geo}} = 384$ to ensure fair comparison.

**DINOv3-based semantic feature extraction:** Each 3DUS volume $I \in \mathbb{R}^{D \times H \times M}$ (with $D = H = M = 64$) is split into its $H \times M$ slices along the sagittal axis, and DINOv3 features are extracted on a per-slice basis. Voxel intensities are linearly normalized to $[0, 255]$, and each single-channel slice is replicated across three channels to form pseudo-RGB inputs. The slices are upsampled by a factor of $s = 16$, corresponding to the ViT patch size, to ensure spatial alignment between the output feature maps and the input images. The resulting inputs are standardized using ImageNet mean and standard deviation and fed into a pretrained DINOv3 ViT-L encoder. Finally, patch tokens from the last layer are reshaped to produce feature maps $F_{\text{Dense}} \in \mathbb{R}^{H \times M \times C}$ for each slice, where $C = 1024$.

For each stitching pair $(I_{\text{source}}, I_{\text{target}})$, dense feature maps from all slices of both volumes are aggregated and projected into a shared low-dimensional embedding space via PCA. This yields PCA-compressed feature maps $F_{\text{PCA}} \in \mathbb{R}^{H \times M \times d_{\text{DINO}}}$ for each 2D slice, where $d_{\text{DINO}} = 16$. For each volume, the compressed feature maps are then stacked along the slice dimension to form a 4D feature volume $V \in \mathbb{R}^{D \times H \times M \times d_{\text{DINO}}}$, which is spatially aligned with the original voxel grid for subsequent registration and analysis.

**Mapping DINOv3 features to PCD:** Next, DINOv3 descriptor $\mathbf{f}_i^{\text{DINO}} \in \mathbb{R}^{d_{\text{DINO}}}$ for each $\mathbf{p}_i$ are extracted by sampling $V$ in the voxel space.

We perform trilinear interpolation over the eight neighboring grid points of $p_i$, which yields $\mathbf{f}_i^{\text{DINO}}$ for each $\mathbf{p}_i$. Collecting all descriptors, we obtain a DINOv3 descriptor set $F_{\text{DINO}} \in \mathbb{R}^{N \times d_{\text{DINO}}}$ that mapping the DINOv3 feature $V$ from voxel space to PCD space.

**Geometric-semantic feature fusion:** For each downsampled point $\mathbf{p}_i$, we fuse its local geometric descriptor $\mathbf{f}_i^{\text{Geo}}$ with the corresponding semantic descriptor $\mathbf{f}_i^{\text{DINO}}$. We first apply row-wise $\ell_2$-normalization to each modality,

$$\tilde{\mathbf{f}}_i^{\text{Geo}} = \frac{\mathbf{f}_i^{\text{Geo}}}{\|\mathbf{f}_i^{\text{Geo}}\|_2 + \varepsilon}, \quad \tilde{\mathbf{f}}_i^{\text{DINO}} = \frac{\mathbf{f}_i^{\text{DINO}}}{\|\mathbf{f}_i^{\text{DINO}}\|_2 + \varepsilon},$$

with a small constant $\varepsilon = 10^{-8}$ for numerical stability. We then form a weighted concatenation of geometric and DINO descriptors,

$$\mathbf{f}_i^{\text{fuse}} = \left[ \alpha \tilde{\mathbf{f}}_i^{\text{Geo}} ; \beta \tilde{\mathbf{f}}_i^{\text{DINO}} \right] \in \mathbb{R}^{d_{\text{Geo}} + d_{\text{DINO}}},$$

where $\alpha, \beta > 0$ control the relative contributions of geometric structure and semantic appearance (set to $\alpha = \beta = 1.0$ by default). Finally, we perform a global $\ell_2$-normalization on the fused descriptors,

$$\hat{\mathbf{f}}_i^{\text{fuse}} = \frac{\mathbf{f}_i^{\text{fuse}}}{\|\mathbf{f}_i^{\text{fuse}}\|_2 + \varepsilon}.$$

These normalized fused descriptors combining complementary geometric and semantic are then used as input to the PCD-based registration.

## 2.3. Registration

**PCD-based registration:** We adopt two widely used robust PCD registration algorithms, Random Sample Consensus (RANSAC) (Fischler and Bolles, 1981) and TEASER++ (Yang et al., 2020), to estimate rigid transformations between 3DUS point clouds.

For RANSAC-based registration, an initial global alignment is obtained through descriptor-based feature matching, with geometric consistency enforced using both edge-length preservation (threshold: 0.9) and Euclidean distance constraints (threshold: 10.0). RANSAC is executed for up to $4 \times 10^6$ iterations, with early termination triggered after 500 inlier correspondences are identified. To further improve local alignment accuracy, the resulting transformation is refined using Iterative Closest Point (ICP) (Besl and McKay, 1992), which minimizes the point-to-point distances between the aligned point clouds.

For TEASER++, we perform robust global rigid registration by directly estimating the rotation $\mathbf{R} \in \text{SO}(3)$ and translation $\mathbf{t} \in \mathbb{R}^3$ between the source and target point clouds. Throughout this work, we denote the instantiations of the proposed framework using RANSAC and TEASER++ as VFMStitch-R and VFMStitch-T, respectively.

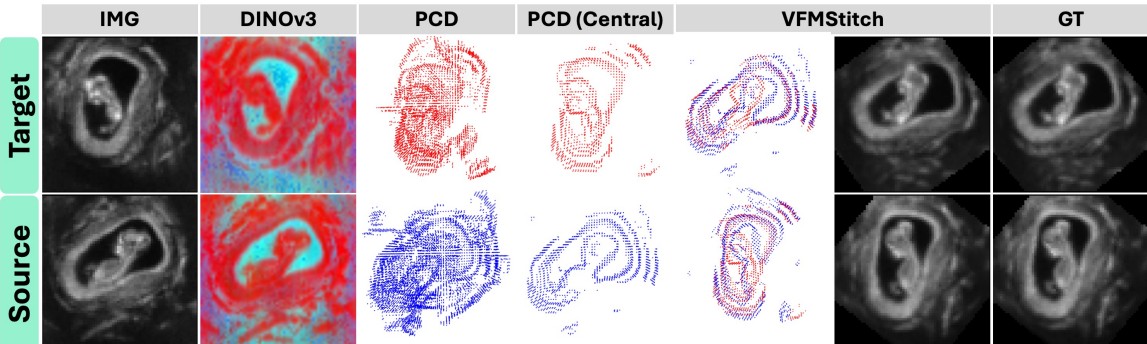

Figure 2: Visualization of the intermediate steps of VFMStitch in the axial plane. The "PCD (Central)" panel shows the extracted point clouds from the central axial slice only for clarity. The first row illustrates registration from target to source, while the second row shows registration from source to target, demonstrating the bidirectional alignment capability of VFMStitch.

**Transformation propagation to volumetric space:** The resulting rigid transformation is represented as a homogeneous matrix $\mathbf{T} \in \mathbb{R}^{4\times4}$ and subsequently applied to the corresponding gray-scale 3DUS volumes. Fig. 2 visualizes the intermediate steps of VFMStitch for a randomly selected example.

**Intensity-based refinement:** We observe that the strong rigid initialization provided by VFMStitch facilitates subsequent intensity-based optimization, leading to further improvements in pixel-level alignment accuracy. To this end, we apply an intensity-based rigid refinement step using ANTs after VFMStitch. This refinement preserves the global alignment established by VFMStitch while exploiting local intensity consistency to further enhance registration accuracy.

### 2.4. Datasets and Evaluation Metrics

**Dataset:** We evaluate the proposed method on the in-house *RegUS* dataset, which comprises 3DUS placenta scans from 20 first-trimester pregnancies. For each subject, two partially overlapping 3DUS volumes are acquired from different probe positions. All volumes are resampled to an isotropic spatial resolution of $(2 \text{ mm})^3$, centrally cropped to $64^3$ voxels, and intensity-normalized to the range $[0,1]$.

**Ground truth:** Ground-truth rigid transformations are manually created by two experienced experts and further visually inspected by three additional reviewers, serving as the reference standard for quantitative evaluation. The manual transformations involve large relative motions, with rotations ranging from $30°$ to $117°$ and translations spanning $[25, 83]$ mm, reflecting realistic and challenging clinical stitching scenarios. Each subject's volumes are registered bidirectionally, resulting in a total of 40 registration pairs.

**Evaluation metrics:** Registration performance is quantitatively assessed using standard image similarity metrics, including normalized cross-correlation (NCC) (Lewis, 1995), struc-

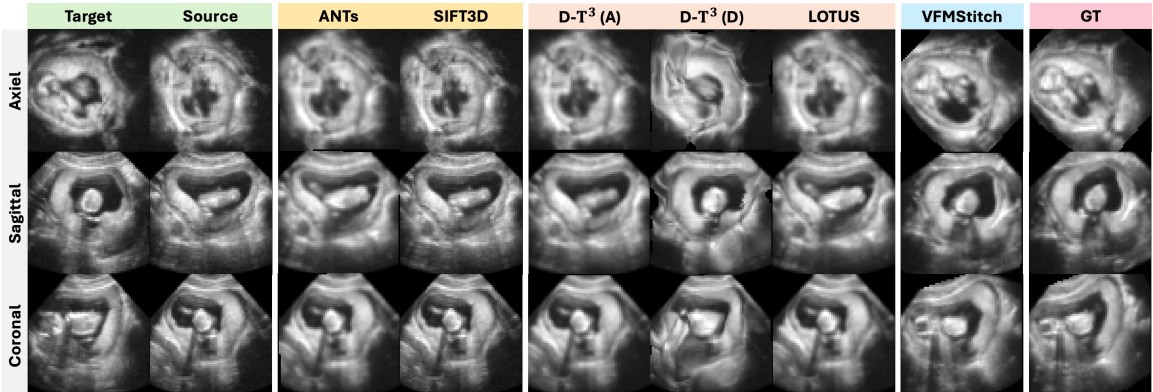

Figure 3: Qualitative comparison on a randomly selected case. D-T³(A) and D-T³(D) denote the rigid and deformable variants of DINOv3-T³, respectively. While the baseline methods fail to achieve satisfactory alignment under large inter-volume motion, VFMStitch produces accurate registration. Notably, D-T³(D) introduces unrealistic geometric distortions, as expected, highlighting the limitations of existing VFM-based deformable registration frameworks in handling large-motion 3DUS stitching.

tural similarity index measure (SSIM) (Wang et al., 2004), peak signal-to-noise ratio (PSNR) (Gonzalez and Woods, 2008), and mean squared error (MSE).

**Implementation details:** We comprehensively evaluate the proposed VFMStitch by comparing it with a diverse set of representative baselines, covering conventional (ANTs (Avants et al., 2008), SIFT3D (Rister et al., 2017)), outpainting-based (LOTUS(Yao et al., 2025a)) and learning-based baselines (ConvexAdam(Siebert et al., 2024), DINOv3+T³(Wang et al., 2025)). For ANTs, we perform rigid registration on both gray-scale 3DUS volumes (denoted as ANTs), and the extracted 4D DINOv3 feature maps (denoted as ANTs-DINO). For LOTUS, ANTs is employed as the registration method following the outpainting. ConvexAdam is evaluated under rigid registration settings using MIND (Heinrich et al., 2012) descriptors. Similarly, DINOv3+T³ is also evaluated in rigid configurations to ensure a fair assessment of its performance under large-motion stitching scenarios, and we also demonstrate its deformable (original setting) registration performance qualitatively. We expect that deformable registration is not suitable to capture rigid transformation and will introduce distortions. All baseline methods are configured according to their recommended settings, and no additional fine-tuning is performed beyond what is required for fair comparison.

## 3. Results and Discussion

**Effectiveness of VFMStitch:** Fig. 3 presents qualitative registration results on a randomly selected example. Conventional methods and learning-based baselines consistently fail due to the strong local minima induced by the sector-shaped FOV. In contrast, VFM-

Table 1: Comparison of VFMStitch and baseline methods on the RegUS dataset. R: RANSAC matcher; T: TEASER++ matcher; F: ANTs-based refinement. Gray: conventional methods; blue: learning-based methods; green: VFMStitch (before refinement); yellow: VFMStitch (after refinement). **Best** and second best results are highlighted. VFMStitch consistently outperforms all baselines across all metrics, with ANTs refinement providing additional gains.

| Methods | NCC ↑ | MSE(×10) ↓ | PSNR ↑ | SSIM ↑ |
|---|---|---|---|---|
| ANTs | $0.826 \pm 0.107$ | $0.174 \pm 0.096$ | $19.894 \pm 6.673$ | $0.496 \pm 0.219$ |
| ANTs-DINOv3 | $0.767 \pm 0.090$ | $0.224 \pm 0.069$ | $16.722 \pm 1.400$ | $0.388 \pm 0.072$ |
| SIFT3D | $0.757 \pm 0.124$ | $0.219 \pm 0.074$ | $16.893 \pm 1.736$ | $0.403 \pm 0.126$ |
| ConvexAdam | $0.773 \pm 0.092$ | $0.218 \pm 0.064$ | $16.813 \pm 1.368$ | $0.391 \pm 0.078$ |
| DINOv3+T$^3$ | $0.778 \pm 0.083$ | $0.217 \pm 0.064$ | $16.829 \pm 1.362$ | $0.390 \pm 0.076$ |
| LOTUS | $0.881 \pm 0.132$ | $0.120 \pm 0.110$ | $21.783 \pm 5.140$ | $0.609 \pm 0.251$ |
| VFMStitch(DINOv3, R) | $0.885 \pm 0.127$ | $0.116 \pm 0.095$ | $20.923 \pm 3.957$ | $0.598 \pm 0.198$ |
| VFMStitch(DINOv3, T) | $0.874 \pm 0.144$ | $0.116 \pm 0.097$ | $21.221 \pm 4.390$ | $0.610 \pm 0.209$ |
| VFMStitch(DINOv3, R, F) | $\mathbf{0.910 \pm 0.114}$ | $\mathbf{0.090 \pm 0.097}$ | $\mathbf{24.064 \pm 6.542}$ | $\mathbf{0.690 \pm 0.242}$ |
| VFMStitch(DINOv3, T, F) | $\underline{0.891 \pm 0.135}$ | $\underline{0.098 \pm 0.094}$ | $\underline{23.260 \pm 6.179}$ | $\underline{0.666 \pm 0.240}$ |

Stitch achieves accurate and visually coherent rigid alignment, effectively preserving anatomical structures across stitched volumes.

Notably, the results further reveal that existing VFM-based registration methods are inadequate for 3DUS stitching when substantial motion is present. Specifically, deformable registration using DINOv3-T$^3$ (D-T$^3$(D)) relies on dense deformation fields to establish pixel-wise correspondences, resulting in unrealistic anatomical distortions, as can be expected. Conversely, the rigid variant (D-T$^3$(A)) fails to recover the correct rigid alignment, highlighting the inherent difficulty of estimating large rotations and translations within a deformation-field optimization framework. These observations indicate a fundamental mismatch between the current VFM-based registration methods and the large motions often present in 3DUS stitching tasks.

Table 1 reports a comprehensive quantitative comparison across all methods. Among existing baselines, LOTUS achieves the strongest overall performance, serving as a competitive state-of-the-art reference. However, the proposed VFMStitch with DINOv3 as descriptor consistently outperforms LOTUS across all evaluation metrics under different PCD registration strategies (RANSAC and TEASER++), with the sole exception of a slightly lower NCC score for VFMStitch-T (DINOv3). Overall, both qualitative and quantitative results demonstrate that VFMStitch provides a more robust and accurate solution for large-motion 3DUS stitching.

**Effectiveness of intensity-based refinement:** We further evaluate the effectiveness of the intensity-based refinement applied after VFMStitch. As reported in Table 1, ANTs-based refinement consistently improves pixel-wise alignment as measured by NCC.

Fig. 4 presents a qualitative comparison before and after refinement. While VFMStitch alone already achieves good global alignment, the subsequent intensity-based refinement

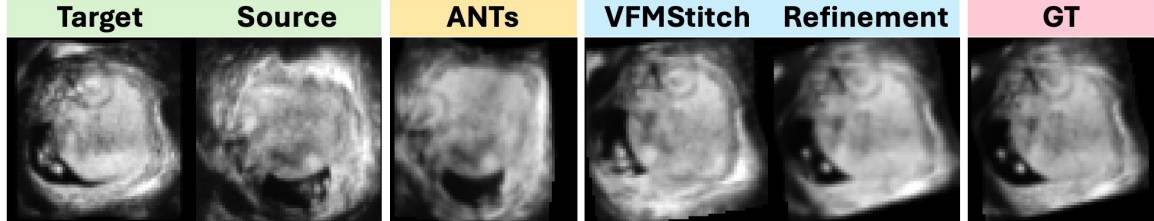

Figure 4: Comparison of registration performance before and after intensity-based refinement using ANTs (Rigid) for VFMStitch (DINOv3+FPFH, T). VFMStitch provides accurate global alignment, which can be further refined by ANTs to improve local registration accuracy. In contrast, ANTs (Rigid) alone fails to achieve satisfactory alignment, highlighting the importance of VFMStitch-based initialization.

further enhances local voxel correspondence. Importantly, the failure of ANTs when applied independently highlights that this improvement critically depends on the strong global initialization provided by VFMStitch. Purely intensity-based registration struggles under large inter-volume motion, whereas its effectiveness is recovered when guided by a robust VFMStitch-based initial alignment. These results confirm that intensity-based refinement serves as a complementary post-processing step, improving fine-scale alignment without altering the robustness or training-free nature of the proposed framework.

**Effectiveness of feature fusion:** Table 2 compares the performance of geometric descriptors alone to fusing geometric and semantic features. We observe that VFMStitch equipped with fused descriptors (red) consistently outperforms both the semantic-only configuration (DINOv3, green) and purely geometric baselines (gray) with the exception of a slight decrease in NCC when fusing DINOv3 with FPFH or Point-DAE. Importantly, this slight reduction does not affect structural similarity or overall registration accuracy, as reflected by the consistent gains in SSIM, PSNR, and MSE. These results suggest that combining low-level geometric structure with high-level semantic context yields more robust and discriminative point correspondences, ultimately improving registration performance in challenging large-motion 3DUS stitching scenarios.

**Effectiveness of DINOv3-based descriptor:** Fig. 5 presents a quantitative comparison between DINOv3-based semantic descriptors and baseline geometric descriptors using different PCD matchers. Across both matching strategies, the DINOv3-based purely semantic descriptor consistently yields superior performance compared to purely geometric descriptors. This trend is consistent across all evaluated metrics, indicating that DINOv3-derived features provide more discriminative and robust representations for establishing reliable correspondences under large rigid motion. We attribute this advantage to the semantic-rich and spatially coherent representations learned by DINOv3 through large-scale self-supervised pretraining, which are better suited to capturing high-level anatomical context than descriptors optimized solely for local geometric patterns. These results demonstrate that DINOv3-derived semantic features can substantially enhance registration robustness in challenging 3DUS stitching scenarios.

Table 2: Comparison of geometric descriptor performance before (gray) and after fusion with DINOv3 semantic features (red), with DINOv3-only results shown in green. RANSAC and ANTs are used for matching and refinement, respectively. **Best** and second best results are highlighted. Geometric–semantic fusion consistently outperforms both geometric-only and semantic-only representations, demonstrating the effectiveness of the proposed fusion strategy.

| Methods | NCC ↑ | MSE(×10) ↓ | PSNR ↑ | SSIM ↑ |
|---|---|---|---|---|
| FPFH (R, F) | $0.889 \pm 0.125$ | $0.105 \pm 0.098$ | $23.256 \pm 6.602$ | $0.654 \pm 0.262$ |
| Point-BERT (R, F) | $0.871 \pm 0.123$ | $0.131 \pm 0.110$ | $21.910 \pm 6.526$ | $0.605 \pm 0.251$ |
| Point-DAE (R, F) | $0.868 \pm 0.122$ | $0.131 \pm 0.106$ | $21.397 \pm 5.693$ | $0.597 \pm 0.244$ |
| Point-MAE (R, F) | $0.866 \pm 0.120$ | $0.137 \pm 0.110$ | $21.310 \pm 5.805$ | $0.592 \pm 0.245$ |
| VFMStitch(DINOv3, R, F) | $\underline{0.910 \pm 0.114}$ | $0.090 \pm 0.097$ | $24.064 \pm 6.542$ | $0.690 \pm 0.242$ |
| VFMStitch(DINOv3+FPFH, R, F) | $0.906 \pm 0.153$ | $0.087 \pm 0.113$ | $24.346 \pm 6.224$ | $\underline{0.709 \pm 0.243}$ |
| VFMStitch(DINOv3+Point-BERT, R, F) | $0.910 \pm 0.126$ | $0.087 \pm 0.099$ | $\mathbf{24.779 \pm 6.932}$ | $0.708 \pm 0.248$ |
| VFMStitch(DINOv3+Point-DAE, R, F) | $0.908 \pm 0.125$ | $\underline{0.085 \pm 0.092}$ | $24.474 \pm 6.581$ | $0.705 \pm 0.243$ |
| VFMStitch(DINOv3+Point-MAE, R, F) | $\mathbf{0.916 \pm 0.116}$ | $\mathbf{0.084 \pm 0.095}$ | $\underline{24.656 \pm 6.586}$ | $\mathbf{0.712 \pm 0.241}$ |

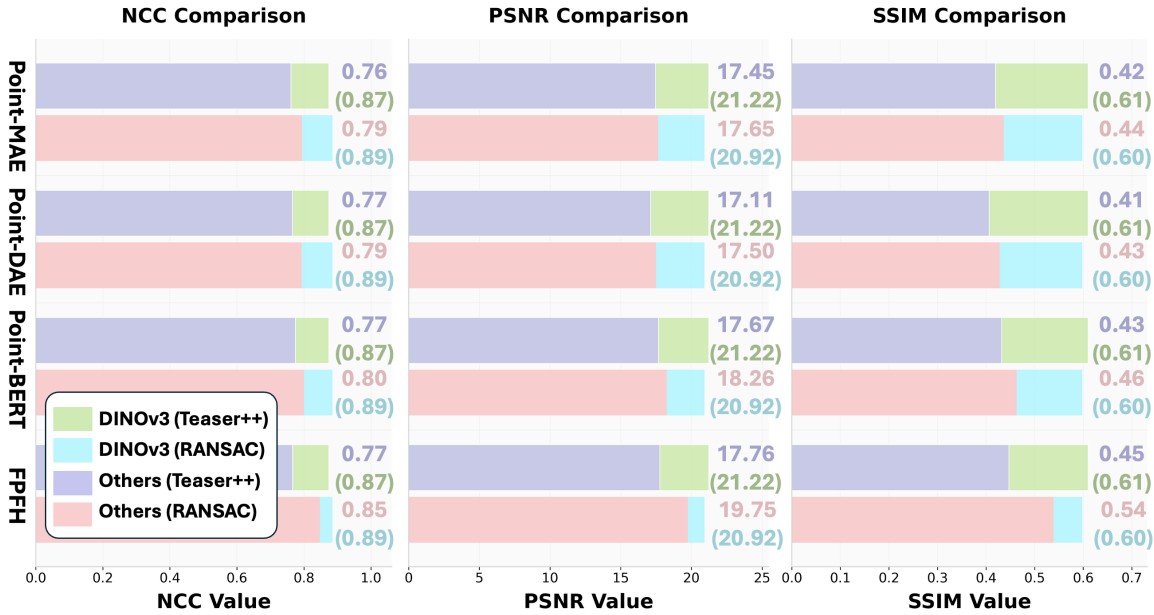

Figure 5: Comparison between DINOv3 and the geometric descriptors using RANSAC and TEASER++ as PCD matchers. Across all evaluated metrics, DINOv3 consistently achieves superior performance, demonstrating its effectiveness as a semantic descriptor for robust point-cloud registration.

**Computational efficiency:** The runtime of VFMStitch is dominated by DINOv3 feature extraction (2.8 s per pair, 2.5 GB memory), PCA (9.5 s), and RANSAC (5.3 s), with FPFH,

Table 3: Ablation study on the impact of PCA dimension $s$ on VFMStitch (DINOv3, R) performance. **Best** and second best results are highlighted. $s = 16$ is the dimension used in all our other experiments, highlighted in gray. Performance remains largely stable across different dimensions, suggesting a robust operating regime rather than dependence on a finely tuned choice.

| PCA Dimension | NCC ↑ | MSE(×10) ↓ | PSNR ↑ | SSIM ↑ |
|---|---|---|---|---|
| $s = 8$ | $0.863 \pm 0.139$ | $0.137 \pm 0.098$ | $20.063 \pm 3.937$ | $0.557 \pm 0.199$ |
| $s = 12$ | $0.891 \pm 0.123$ | $0.111 \pm 0.089$ | $21.168 \pm 4.015$ | $0.603 \pm 0.208$ |
| $s = 16$ | $0.885 \pm 0.127$ | $0.116 \pm 0.095$ | $20.923 \pm 3.957$ | $0.598 \pm 0.198$ |
| $s = 20$ | $0.895 \pm 0.122$ | $0.108 \pm 0.089$ | **$21.240 \pm 3.901$** | **$0.610 \pm 0.202$** |
| $s = 24$ | **$0.896 \pm 0.119$** | **$0.108 \pm 0.087$** | $21.217 \pm 3.892$ | $0.609 \pm 0.199$ |

ICP, and Teaser++ each contributing less than 0.1 s. Overall, the core components of VFMStitch require about 18s per pair without ANTs refinement. This is notably more efficient than diffusion-based stitching methods such as LOTUS, which requires approximately 29s per pair at inference time and over 20GB memory during training. While VFMStitch is slower than ANTs (2.5s per pair), it improves accuracy substantially without any training or fine-tuning cost, making it suitable for offline clinical use.

**Robustness to PCA dimension:** Table 3 demonstrates the sensitivity of VFMStitch (DINOv3, R) to the PCA dimension $s$ used for semantic feature compression. Performance remains stable across a broad range ($s \in [12, 24]$), with only marginal small variations, indicating a clear performance plateau. Minor improvements for certain $s$ values fall within this plateau, while degradation occurs only at very low dimensionality ($s = 8$) due to limited semantic capacity rather than framework instability.

**Robustness to fusion weights:** Table 4 shows the sensitivity of VFMStitch (DINOv3 + FPFH, R) to the fusion weights $(\alpha, \beta)$ controlling the contribution of semantic and geometric features. Performance remains largely stable across a broad range of weight combinations.

**Robustness to sampling ratio:** Appendix Fig. A1 shows the effect of the sampling ratio $r$ on stitching performance. As the $r$ decreases, all methods consistently benefit from finer geometric and semantic discretization, resulting in improved registration accuracy across all metrics. Importantly, the relative performance ranking remains unchanged, with the VFMStitch(DINOv3+FPFH, R) and VFMStitch(DINOv3, R) consistently outperforming baseline FPFH across all sampling factors. Notably, at $r = 1$, VFMStitch(DINOv3+FPFH, R) achieves 0.96 for NCC even without ANTs refinement. We also observe that VFMStitch is more robust than FPFH to inadequate sampling. For example, when the sampling ratio increases from $r = 1$ to $r = 9$, the NCC of FPFH drops by 15%, whereas VFMStitch (DINOv3, R) and VFMStitch (DINOv3+FPFH, R) only drop by 7% and 8%, respectively. These results show that sampling factor $r$ affects accuracy, while also highlighting the robustness and effectiveness of the proposed geometric–semantic fusion strategy.

**Overlap robustness and failure cases:** We report the effect of overlap between the stitching pair on performance in Appendix B, and analyze a failure case in Appendix C.

Table 4: Ablation study on the fusion weights $\alpha$ and $\beta$ on VFMStitch (DINOv3+FPFH, R) performance. **Best** and second best results are highlighted. $\alpha = 1, \beta = 1$ are the weights used in all our other experiments, highlighted in gray. Performance remains largely stable across different fusion weights, suggesting a robust operating regime rather than dependence on a finely tuned choice.

| Fusion Weights | NCC ↑ | MSE(×10) ↓ | PSNR ↑ | SSIM ↑ |
|---|---|---|---|---|
| $\alpha = 0.6, \beta = 1.0$ | $0.891 \pm 0.124$ | $0.111 \pm 0.091$ | $21.132 \pm 3.996$ | $0.608 \pm 0.197$ |
| $\alpha = 0.7, \beta = 1.0$ | $0.875 \pm 0.141$ | $0.122 \pm 0.088$ | $20.577 \pm 3.981$ | $0.575 \pm 0.202$ |
| $\alpha = 0.8, \beta = 1.0$ | $0.885 \pm 0.142$ | $0.116 \pm 0.090$ | $20.702 \pm 3.618$ | $0.593 \pm 0.191$ |
| $\alpha = 0.9, \beta = 1.0$ | $\mathbf{0.908 \pm 0.128}$ | $\mathbf{0.097 \pm 0.076}$ | $\mathbf{21.457 \pm 3.642}$ | $\mathbf{0.628 \pm 0.178}$ |
| $\alpha = 1.0, \beta = 1.0$ | $0.884 \pm 0.148$ | $0.115 \pm 0.098$ | $21.031 \pm 3.986$ | $0.602 \pm 0.201$ |
| $\alpha = 1.0, \beta = 0.9$ | $0.895 \pm 0.140$ | $0.106 \pm 0.088$ | $21.259 \pm 3.773$ | $0.615 \pm 0.193$ |
| $\alpha = 1.0, \beta = 0.8$ | $0.890 \pm 0.143$ | $0.112 \pm 0.095$ | $21.012 \pm 3.830$ | $0.605 \pm 0.192$ |
| $\alpha = 1.0, \beta = 0.7$ | $0.899 \pm 0.109$ | $0.109 \pm 0.088$ | $21.115 \pm 3.841$ | $0.610 \pm 0.195$ |
| $\alpha = 1.0, \beta = 0.6$ | $0.878 \pm 0.148$ | $0.119 \pm 0.093$ | $20.704 \pm 3.885$ | $0.588 \pm 0.199$ |

**Discussion and conclusion:** In this work, we presented **VFMStitch**, the first training-free framework that systematically integrates vision foundation model–derived semantic features with PCD geometric registration for large-motion 3DUS stitching. By explicitly reformulating 3DUS stitching as a robust rigid registration problem in the PCD domain, VFM-Stitch overcomes fundamental limitations of existing VFM-based approaches under large translations and rotations. Extensive quantitative and qualitative evaluations demonstrate that VFMStitch consistently outperforms state-of-the-art conventional, learning-based, and VFM-based baselines, highlighting its effectiveness for challenging ultrasound data.

Beyond overall performance, our ablation and sensitivity analyses indicate that VFM-Stitch operates in a robust regime across a broad range of design choices, without reliance on finely tuned hyperparameters. This suggests that the proposed framework is not specific to a particular dataset configuration, and is expected to maintain stable behavior in larger datasets.

While the current evaluation focuses on placental 3DUS data, the core design of VFM-Stitch is anatomy- and protocol-agnostic, as it relies on generic geometric constraints and pretrained foundation-model features rather than organ-specific supervision. We therefore anticipate that the proposed framework can be extended to other anatomical targets and ultrasound acquisition protocols, in the context of similar rigid or near-rigid medical image registration tasks where large motion and low image quality pose challenges to traditional intensity-based methods. These further validation experiments remain as future work.

Overall, this work demonstrates the potential of training-free, foundation-model-guided geometric registration for challenging medical imaging scenarios, and opens new directions for leveraging vision foundation models beyond conventional intensity-driven pipelines.

## Acknowledgments

This work is supported, in part, by NIH R01-HD109739, R01-HL156034, T32-EB021937, and the Vanderbilt Advanced Computing Center for Research and Education.

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

## Appendix A.  Voxel sampling factor robustness

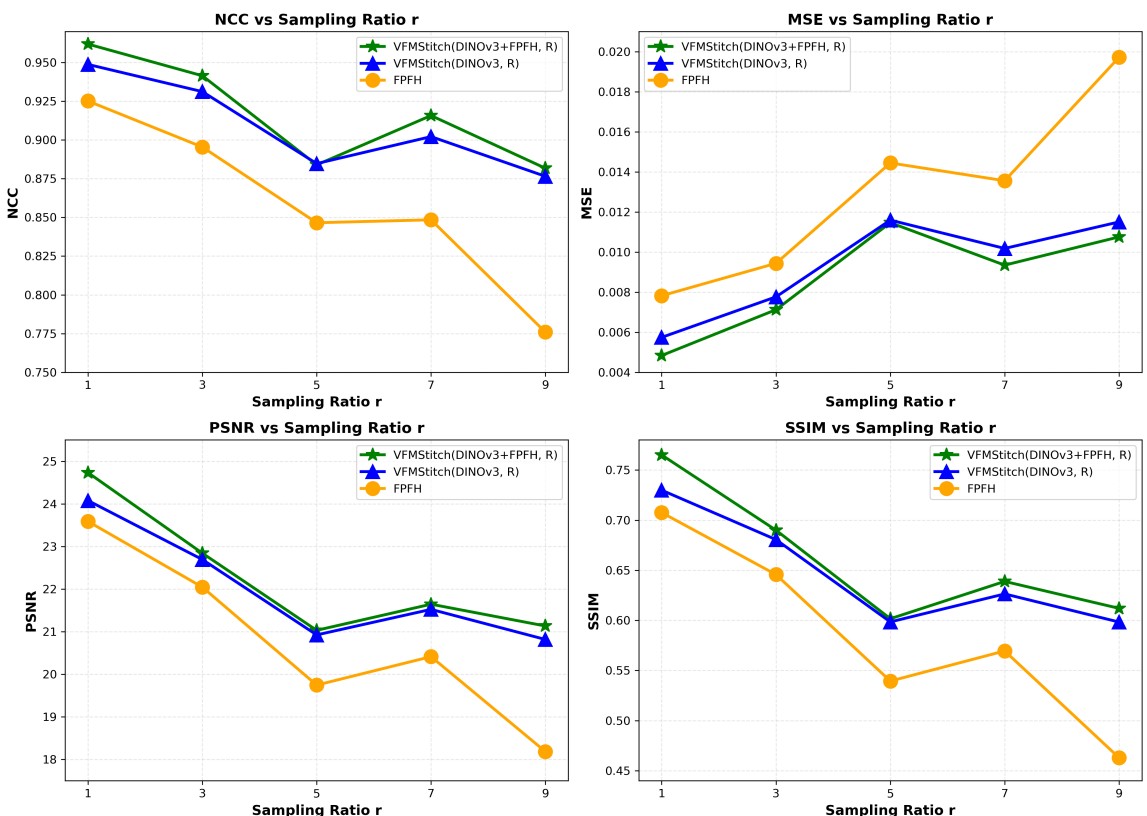

Figure A1: Performance under different voxel sampling factors $r$ measured by NCC, MSE, PSNR and SSIM. VFMStitch(DINOv3+FPFH, R), VFMStitch(DINOv3, R) and FPFH are compared. Note that the default setting for VFMStitch used throughout the other experiments is $r = 5$. Decreasing $r$ consistently improves performance for all methods, but the performance ranking of the methods remains stable.

## Appendix B.  Overlap robustness

To quantitatively assess the sensitivity of VFMStitch to varying degrees of inter-volume overlap, we estimate the overlap ratio for each image pair. Specifically, the Dice coefficient (DSC) is computed between the sector-shaped FOVs of the manually registered ground-truth volume and the corresponding fixed image, serving as a proxy for the amount of shared anatomical content between the two volumes.

In Fig.A2, we report the correlation between overlap ratio (DSC) and registration accuracy measured by normalized cross-correlation (NCC) under different sampling ratios. We do not observe a strong correlation between overlap ratio and registration accuracy. This

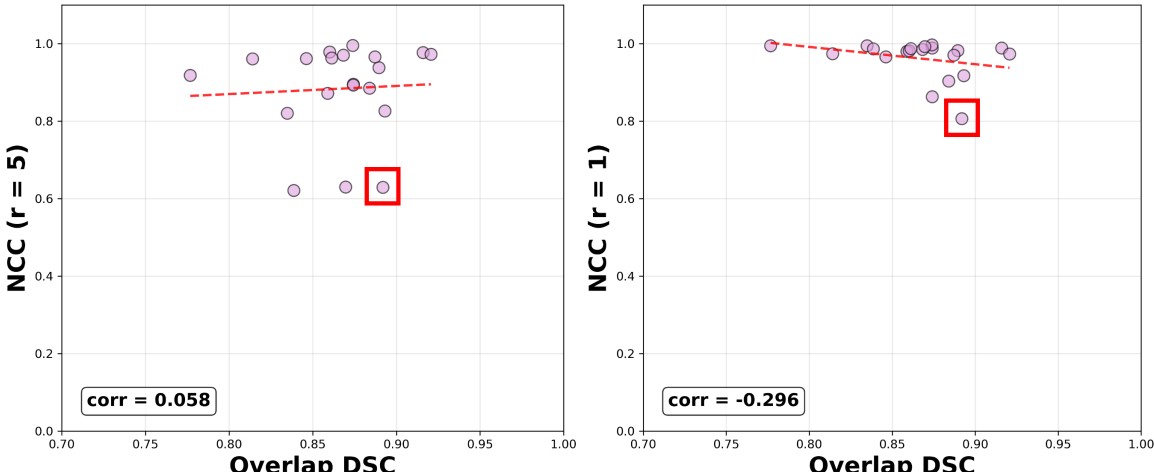

Figure A2: Correlation between overlap ratio (DSC) and stitching performance (NCC) for two different sampling ratios ($r = 5$ and $r = 1$). The red dashed lines represent linear trend fits, with Pearson correlation coefficients indicated in the legend. Results show weak correlations (0.058 for $r = 5$ and 0.296 for $r = 1$), suggesting that the stitching quality is relatively stable across overlap ratios in this overlap range. The red box highlights a challenging stitching pair further analyzed in Appendix C.

suggests that as long as there is adequate overlap for stitching, the precise overlap amount is not a meaningful factor for registration performance.

## Appendix C. Failure case analysis

To further investigate the robustness of VFMStitch under particularly challenging conditions characterized by the sum of absolute rotations along the three axes exceeding $120°$, we analyze a representative failure case, as shown in Fig. A3. This stitching pair is also corresponding to the sample highlighted by the red highlight box in Fig. A2. This example represents one of the most difficult registration scenarios in the dataset with large rotation.

Under the default sampling ratio ($r = 5$), all stitching methods fail to recover a correct alignment on this pair. In contrast, with denser sampling ($r = 1$), substantially more geometric and semantic detail is preserved in the point-cloud representation. Under this setting, VFMStitch successfully recovers the correct alignment for this challenging pair (Fig. A3).

We observe that this denser sampling ($r = 1$) similarly yields consistent performance improvements across nearly all image pairs in the dataset. The result indicates that the observed failure is not due to an inherent limitation of the proposed framework, but rather arises from the interaction between extreme motion and coarse sampling during point-cloud construction. This observation highlights the importance of adequate PCD sampling for

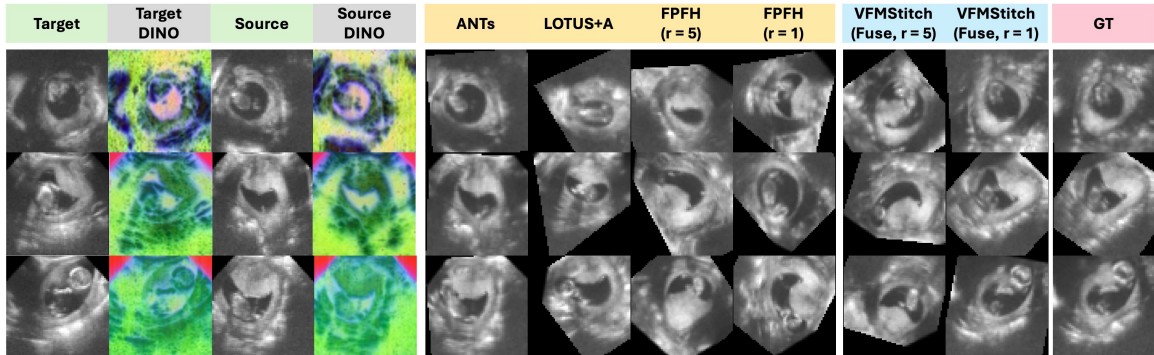

Figure A3: Qualitative results on a challenging stitching pair with severe rotation, with the sum of rotations along the three axes exceeding 120°. With the default sampling ratio ($r = 5$), all methods produce incorrect alignments. With denser sampling ($r = 1$), VFMStitch is able to recover the correct alignment wheras FPFH still fails.

handling extreme motion and further demonstrates that the proposed geometric–semantic fusion strategy remains effective when sufficient spatial information is available.

