# OpenReview forum: "VFMStitch: A Vision-Foundation-Model Empowered Framework for 3D Ultrasound Stitching via Geometric–Semantic Feature Fusion"
_MIDL.io/2026/Conference — MIDL 2026 Poster_

### Official Review · Reviewer_qT3R · 2025-12-28

**Confidence:** 3
**Preliminary Rating:** 4
**Final Rating:** 4

**Summary:**

This paper presents a training-free framework (VFMStitch) for rigid 3D ultrasound stitching under large inter-volume motion. The proposed method leverages DINOv3 vision foundation model features as semantic descriptors and fuses them with point-cloud–based geometric features extracted via unsupervised edge detection, enabling robust point-cloud registration using RANSAC or TEASER++. The estimated rigid transformation is refined with intensity-based registration (ANTs). Experiments on a private dataset demonstrate that VFMStitch consistently outperforms conventional learning-based and recent VFM-based baselines across multiple metrics, particularly in challenging large-motion scenarios. The work highlights the effectiveness of geometric–semantic feature fusion and provides evidence that VFM-derived features generalize well to noisy, artifact-prone ultrasound data without additional training.

**Strengths:**

- This work addresses an important and practically relevant problem in medical imaging on robust rigid registration for 3D ultrasound stitching, where large motion, low signal-to-noise ratio, and sector-shaped fields of view make traditional intensity-based methods unreliable.
- The key contribution is the training-free design, which avoids the data-hungry and modality-specific training requirements of recent learning-based stitching approaches and improves robustness across acquisition variability. The integration of DINOv3 semantic descriptors with point-cloud geometric features is well motivated and empirically validated.
- Ablation studies and experimental evaluation are thorough for the given dataset, with clear qualitative visualizations and consistent improvements across multiple metrics and registration strategies.
- Overall, the method is well motivated and straightforward, and demonstrates strong empirical performance for a difficult ultrasound registration setting.
- The presentation is clear and reads well.

**Weaknesses:**

- The methodological novelty is primarily incremental, relying on existing components (edge-based PCD extraction, DINOv3 features, RANSAC/TEASER++, ANTs refinement) assembled into an overall pipeline.
- The evaluation is limited to a single in-house dataset (20 subjects, 40 registration pairs), which raises concerns about generalization to other data and domain thave have varying anatomical targets, scanners, or acquisition protocols.
- Although intensity-based refinement improves results, it partially blurs the boundary between the proposed method and classical pipelines, making it harder to isolate the unique contribution of VFM-guided registration alone.
- The method seems to assume access to reasonably overlapping volumes, and its performance may degrade when overlap is small or when anatomy is highly repetitive. Failure cases are not explicitly analyzed.

**Detailed Comments:**

- The paper would benefit from a brief discussion of computational complexity and runtime, particularly for different configurations considered.
- Reporting robustness across different overlap ratios or simulated noise levels could strengthen claims of general applicability.

**Justification Of Final Rating:**

Thank you for the detailed response and the additional results on failure cases

My minor concerns have been addressed. I believe this work will be of general interest to the MIDL audience, and I will keep my acceptance recommendation.

**Justification Of The Preliminary Rating:**

This paper offers a well-motivated and practical solution to a critical medical image registration problem. Although the individual components are not novel in isolation, their combination represents a meaningful contribution. The experimental results are consistent and convincing within the evaluated setting, and the approach is likely to be useful to practitioners facing similar challenges. With broader validation and minor clarifications, this work would serve as a solid reference for future research. Therefore, I am leaning towards accepting.

**Questions To Address In The Rebuttal:**

- How well does VFMStitch generalize to other anatomical regions or ultrasound protocols, and are there results on additional, public datasets?
- How sensitive is performance to the degree of overlap between volumes, and are there failure cases where semantic features are unreliable?

---

> ### Author Response · Authors · 2026-01-25
>
> We sincerely appreciate your review and the constructive comments you provided. We provide a point-by-point response below.
>
> - **Novelty:** We agree of course with the reviewer that the individual components are not novel. But, as the reviewer suggests themselves, we believe their combination is a meaningful novel contribution to the field.
>
> - **Generalizability to other datasets:** We thank the reviewer for this important question. Large-motion 3D ultrasound stitching requires paired volumes acquired from the same subject and hardware with overlapping content, and there are no public benchmarks that satisfy this requirement. Nevertheless, VFMStitch is anatomy- and protocol-agnostic by design, as it relies on generic geometric constraints and pretrained foundation-model features rather than organ-specific supervision. While additional public datasets would be valuable for further validation, we plan to extend the evaluation to other anatomical targets and acquisition protocols as future work. We clarify this limitation and discussion in the revised manuscript (p12).
>
> - **Influence of ANTs refinement:** We thank the reviewer for this concern. ANTs is used purely as an optional post-hoc refinement and is not required for the core VFM-guided registration. As shown across multiple experiments (Tables 1, 3, and 4, and Figs. 4–8), VFMStitch already outperforms all baselines without ANTs refinement, while ANTs alone fails under large motion. These results demonstrate the independent contribution of the VFM-guided registration.
>
> - **Robustness across overlap ratios:** We thank the reviewer for raising up this interesting concern. Naturally, the performance is expected to decline with overlap ratio: at the two extremes, full overlap would yield the best performance, whereas no overlap would make stitching impossible. We estimate the overlap ratio of each image pair using the Dice coefficient (DSC) computed between the sector-shaped FOVs of the manually registered ground truth and the fixed image. As shown in Appendix Fig. A2, we analyze the relationship between this overlap ratio (DSC) and the registration accuracy (NCC) under different sampling rates, and report the Pearson correlation coefficients. Since our data was collected specifically with the goal of stitching, the DSC values are within the [0.75, 1] range. We observe weak correlations (< 0.3) between the NCC and DSC in this overlap range, suggesting VFMStitch achieves consistently strong performance across most samples, indicating robustness to the specific overlap ratio as long as there is meaningful overlap. We also agree with the reviewer that in the case of highly repetitive anatomy such as perhaps the spine, the correspondences between the two images may be too ambiguous to resolve without additional context.
>
> - **Computational complexity:** We thank the reviewer for this helpful suggestion. In the revised manuscript (p10), we include a brief discussion of computational complexity together with a component-wise runtime and memory analysis. Specifically, DINOv3 feature extraction takes 2.8s per pair (∼2.5GB memory), while PCA and RANSAC require 9.5s and 5.3s per pair, respectively; FPFH, ICP, and Teaser++ each take less than 0.1s per pair. Overall, the core pipeline of VFMStitch requires approximately 18s per pair without ANTs refinement. This runtime is notably faster than diffusion-based stitching methods such as LOTUS (∼29s per pair at inference), while achieving substantially improved robustness compared to conventional registration approaches such as ANTs.
>
> - **Failure case analysis:** We thank the reviewer for this insightful suggestion. We additionally examine a representative failure case with the sum of rotations along the three axes exceeding 120°, as shown in Appendix Fig. A3. With the default sampling ratio (r=5), all evaluated methods fail on this challenging pair. In contrast, using denser sampling (r=1) allows VFMStitch to successfully recover the alignment, while FPFH still struggles with it. This denser sampling also leads to consistent VFMStitch performance improvements across nearly all cases. This result indicates that the observed failure is primarily driven by the combined effect of extreme motion and coarse sampling, rather than an inherent limitation of the proposed framework.

---

> ### Comment · Reviewer_y5e7 · 2026-01-27
> **concerns addressed**
>
> Thanks for the detailed responses, my concerns regarding runtime and sensitivity analysis have been adequately addressed.

---

### Official Review · Reviewer_9ggt · 2026-01-10

**Confidence:** 4
**Preliminary Rating:** 4
**Final Rating:** 4

**Summary:**

VFMStitch introduces the first training-free, VFM-empowered 3DUS stitching framework that integrates geometric features with DINOv3-derived semantic descriptors to perform robust rigid registration under large inter-volume translations and rotations in challenging 3D ultrasound placenta imaging. Experiments on the RegUS dataset show that VFMStitch consistently outperforms conventional, learning-based, and VFM-based baselines such as ANTs, SIFT3D, ConvexAdam, LOTUS, and DINOv3+test-time training.

**Strengths:**

The proposed approach is novel and the derived conclusions follow well from the performed experiments. Apart from that the specific strengths of the paper are:
1. A training-free stitching framework that performs robust rigid registration under large inter-volume motions without task-specific learning.
2. A segmentation-free point-cloud construction strategy avoiding reliance on organ/structure masks that are costly to obtain and may not generalize.
3. A comprehensive, component-wise evaluation against state-of-the-art conventional, learning-based, and VFM-based baselines (Table 1, Table 2, Fig. 3–4), clearly demonstrating the contribution of geometric–semantic fusion and intensity-based refinement to overall performance.

**Weaknesses:**

1. There are too many design choices in this approach, making this far from elegant. The robustness of these choices and approach in larger datasets is questionable.
2. The central conclusion is that when there is a large displacement such as in 3D US imaging, stitching approach should first focus on rigid transformation followed later by intensity-based refinement that accounts for deformation as well. This approach is shown to be better than doing a deformable registration from the beginning. While this is shown to work in a small dataset (with only 20 image pairs used for stitching), robustness is large datasets remains in question.

**Detailed Comments:**

See the sections above.

**Justification Of Final Rating:**

The authors have adequately addressed my main concerns in their rebuttal and in the revised manuscript. I am now more confident in the robustness and relevance of the work and view it suitable for acceptance. Although the size of the dataset is a cause for concern, this work provides a compelling and well-validated framework.

**Justification Of The Preliminary Rating:**

The use of vision foundational model such as DINOv3 for the purpose of 3D US stitching has been underexplored so far. This paper presents a novel approach to use foundational model for free-hand US stitching. Given the size of the dataset and the number of design choices being made in this approach, the usefulness in larger datasets and actual clinical use remains in question. But this is an important contribution in the field of 3D Ultrasound stitching nonetheless.

**Questions To Address In The Rebuttal:**

VFMStitch relies on multiple design choices: edge-based PCD extraction with Sobel/LoG/Harris, thresholding and 1/5 subsampling, fusion weights, RANSAC thresholds/iterations, ICP refinement, ANTs refinement.
Can you provide a sensitivity analysis showing how performance varies with these key hyperparameters and design choices, to support the robustness of the framework?

---

> ### Author Response · Authors · 2026-01-25
>
> We sincerely appreciate your insightful review and valuable constructive feedback. To thoroughly address your concerns, we have provided a detailed, point-by-point response.
>
>  **Stability on larger dataset:** We thank the reviewer for raising this important question regarding robustness and generalizability. As discussed above in response to R1, there are no suitable large public datasets available for more extensive benchmarking. However, VFMStitch is motivated by low-data, high-variability clinical scenarios, where large-scale annotated datasets are often difficult to obtain in practice, making learning-based methods unreliable. To assess the robustness of our approach, we conducted targeted sensitivity analyses on key parameters, including PCA dimensionality, fusion weights, and volume subsampling ratio, as detailed below. While validation on larger datasets would be valuable, the current results suggest that the proposed framework operates in a robust regime and does not rely on excessive parameter tuning.
>
> **Sensitivity to key parameters:** We have added a careful analysis of our method’s sensitivity to these components, as detailed below. We believe this has substantially improved our manuscript, so we thank the reviewer for the constructive and detailed criticism.
>
> - **(a) Sensitivity to traditional components:** Many of the listed components are intentionally designed to be robust rather than tuned. Edge-based point clouds are constructed by aggregating complementary detectors (Sobel/LoG/Harris) instead of selecting a single operator. Binary thresholding uses a fixed default value (0.5). RANSAC, ICP, and ANTs are each applied with the standard/default settings without dataset-specific optimization. In particular, ANTs is used purely as a post-hoc intensity-based refinement step and is not required for the core geometric–semantic registration. These components therefore do not introduce vulnerability to parameter sensitivity.
>
> - **(b) Sensitivity of sampling ratio:** This was a particularly useful criticism and we further thank the reviewer for it. We conducted a sensitivity study by varying the sampling ratio (p11 and the new Appendix Fig. A1). We observe that decreasing the sampling ratio r from 5 to 1 consistently improves absolute performance for all methods, as denser point clouds preserve more geometric detail and semantic features. Importantly, the VFMStitch outperforms the baseline FPFH for all settings of r. Moreover, VFMStitch is more robust to inadequate sampling than FPFH: from r=1 to r=9, FPFH shows a 15% drop in NCC, while VFMStitch variants only drop by 7% to 8%. Notably, even without ANTs refinement, VFMStitch (DINOv3+FPFH, R) achieves an NCC of up to 0.96 when the sampling ratio is 1, successfully stitching nearly all image pairs. These results indicate that sampling indeed affects accuracy, especially for challenging pairs. They also highlight the robustness and relative effectiveness of the proposed framework compared to the FPFH method.
>
> - **(c) Sensitivity of PCA-dimension and fusion weights:** We conducted ablation studies on PCA dimensionality and fusion weights, as reported in Tables 3 and 4 of the revised manuscript. The results show stable performance across broad operating ranges, with only small variations across all metrics. This indicates that VFMStitch does not rely on fine-grained hyperparameter tuning but operates in a robust regime.

---

### Official Review · Reviewer_y5e7 · 2026-01-16

**Confidence:** 4
**Preliminary Rating:** 4

**Summary:**

This paper introduces VFMStitch, a training-free framework for large-motion 3D ultrasound (3DUS) stitching that integrates vision foundation model (DINOv3) semantic features with point-cloud-based geometric registration. The method reformulates stitching as a robust rigid registration problem under large translations and rotations, followed by optional intensity-based refinement. Experiments on a challenging placenta 3DUS dataset demonstrate consistent improvements over conventional, learning-based, and recent VFM-based baselines across multiple similarity metrics. Overall, the work highlights the effectiveness of geometric–semantic feature fusion and shows that foundation-model features can be successfully leveraged for rigid registration in low-quality ultrasound data.

**Strengths:**

The paper addresses a well-motivated and practically important problem, namely large-motion 3DUS stitching, where many existing methods fail. The proposed framework is methodologically novel in combining DINOv3 semantic descriptors with point-cloud-based rigid registration in a fully training-free manner. The experimental evaluation is thorough, includes strong baselines, and uses clinically realistic motion ranges. The ablation studies convincingly demonstrate the benefit of semantic features and geometric–semantic fusion. The paper is well structured, clearly written, and situates itself appropriately within prior work on ultrasound registration and VFM-based methods.

**Weaknesses:**

The evaluation is limited to a single in-house dataset, which may raise questions about generalizability to other ultrasound anatomies or acquisition protocols. While the method is training-free, it relies on computationally heavy components (e.g., DINOv3 feature extraction and large-scale RANSAC), and runtime or memory analysis is not discussed. The choice of fusion weights and PCA dimensionality is fixed, and sensitivity to these hyperparameters is not explored. Finally, while qualitative comparisons are strong, additional failure case analysis would help clarify remaining limitations.

**Detailed Comments:**

Reporting runtime and memory usage would improve practical relevance.
Clarify how sensitive performance is to PCA dimension and fusion weighting.
A brief discussion on extending the method to other organs or probe geometries would strengthen the impact.
Visualization of failure cases could further contextualize robustness.

**Justification Of The Preliminary Rating:**

The paper presents a clear methodological contribution that fills an important gap in current VFM-based registration research by addressing large rigid motions in noisy 3D ultrasound. The results are strong, consistent, and well supported by both quantitative and qualitative evidence. While generalizability and computational cost warrant further clarification, these limitations do not outweigh the paper’s novelty, technical quality, and relevance to the MIDL community. Overall, this work represents a solid and meaningful contribution.

**Questions To Address In The Rebuttal:**

How does VFMStitch generalize to other 3DUS datasets or anatomical targets beyond placenta imaging?
What is the computational cost compared to leading baselines, and is real-time use feasible?
How sensitive is performance to PCA dimension and fusion weights?

---

> ### Author Response · Authors · 2026-01-25
>
> We genuinely appreciate your thoughtful review and the valuable constructive comments. We have provided a detailed, point-by-point response to address your concerns.
>
> - **Generalizability of VFMStitch:** We thank the reviewer for this important question. Large-motion 3DUS stitching requires paired volumes acquired from the same subject and hardware, with overlapping anatomical content under substantial relative motion. As a result, there is currently no publicly available benchmark for this setting, limiting cross-dataset evaluation. Nevertheless, VFMStitch is anatomy-agnostic by design, as it relies on generic geometric constraints and pretrained foundation-model features rather than organ-specific supervision or training. In ongoing and future work, we plan to collect additional 3D ultrasound data across different anatomical targets and acquisition protocols to further evaluate and extend the generalizability of the proposed framework. We clarify this limitation in the discussion section of the revised manuscript (p12).
>
> - **Computational Cost:** We thank the reviewer for this question regarding computational efficiency. In the revised manuscript (p10), we provide a component-wise runtime and memory analysis. Specifically, DINOv3 feature extraction takes 2.8s per pair with approximately 2.5GB memory usage, while PCA and RANSAC require 9.5s and 5.3s per pair, respectively; FPFH, ICP, and Teaser++ each take less than 0.1s per pair. Overall, the core components of VFMStitch require about 18s per pair without ANTs refinement. This is notably more efficient than diffusion-based stitching methods such as LOTUS, which requires approximately 29s per pair at inference time and over 20GB memory during training. While VFMStitch is slower than ANTs (2.5s per pair), it  improves accuracy substantially without any training or fine-tuning cost, making it suitable for offline clinical use. Real-time stitching is not our current goal, however, further efficiency gains are expected through model distillation and GPU acceleration in future work.
>
> - **Sensitive to PCA dimension and fusion weights:** We thank the reviewer for this insightful question. We have now conducted ablation studies on both PCA dimensionality and fusion weights, as reported in the new Tables 3 and 4 of the revised manuscript. The results show that VFMStitch exhibits stable performance across a broad range of PCA dimensions and fusion weight combinations, with only small variations across all metrics. Although certain settings lead to slightly better metrics, these are within the same performance tier. Overall, the results demonstrate that VFMStitch is robust to the choice of PCA dimension and fusion weights.
>
> - **Visualization of failure cases:** We thank the reviewer for this valuable suggestion. Following this recommendation, we include a failure case analysis on an extremely challenging pair with the sum of absolute rotations along the three axes exceeding 120°, as shown in the new Appendix Fig. A3 and the new Appendix C . We find that the failure arises from a combination of case difficulty and large sampling ratio, rather than limitations of the proposed framework itself. Under the default sampling ratio (r = 5), all methods, including the baselines, fail on this example, whereas denser sampling (r = 1) enables VFMStitch to successfully recover the alignment on this pair and consistently improves performance across nearly all cases.

---

### Author Rebuttal · Authors · 2026-01-25

**Rebuttal:**

We sincerely thank the reviewers for their valuable comments and constructive suggestions, which have helped us improve the manuscript and inspired our future work. We have carefully revised the paper to address all concerns.

**Supporting Material:**

/attachment/feb9175fbb7ab1967cc9be53a482763eeaa37385.pdf

---

### Comment · Area_Chair_iSLU · 2026-01-25
**discussion period is open**

Dear Reviewers,

The authors have submitted their responses to the comments you raised. The paper is now open for discussion.

Please engage with the authors during this period to clarify any remaining issues.

---

### Meta-Review · Area_Chair_iSLU · 2026-02-06

**Recommendation:** Accept (Poster)
**Confidence:** 5

**Metareview:**

All reviewers are in favor of accepting the work, and the main concerns were adequately addressed during the rebuttal.

---

### Decision · Program_Chairs · 2026-02-13

Accept (Poster)